# Acquisition of ionic copper by the bacterial outer membrane protein OprC through a novel binding site

Satya Prathyusha Bhamidimarri[1], Tessa R. Young[2], Muralidharan Shanmugam[3], Sandra Soderholm[4], Arnaud Baslé[1], Dirk Bumann[4], Bert van den Berg[1]*

1 Biosciences Institute, The Medical School, Newcastle University, Newcastle upon Tyne, United Kingdom, 2 Department of Biosciences, Durham University, United Kingdom, 3 Photon Science Institute and Manchester Institute of Biotechnology, University of Manchester, Oxford Road, United Kingdom, 4 Focal Area Infection Biology, University of Basel, Basel, Switzerland

* bert.van-den-berg@ncl.ac.uk

**Data Availability Statement:** All relevant data are within the paper and its Supporting Information files. Coordinates and structure factors have been

## Abstract

Copper, while toxic in excess, is an essential micronutrient in all kingdoms of life due to its essential role in the structure and function of many proteins. Proteins mediating ionic copper import have been characterised in detail for eukaryotes, but much less so for prokaryotes. In particular, it is still unclear whether and how gram-negative bacteria acquire ionic copper. Here, we show that *Pseudomonas aeruginosa* OprC is an outer membrane, TonB-dependent transporter that is conserved in many Proteobacteria and which mediates acquisition of both reduced and oxidised ionic copper via an unprecedented CxxxM-HxM metal binding site. Crystal structures of wild-type and mutant OprC variants with silver and copper suggest that acquisition of Cu(I) occurs via a surface-exposed "methionine track" leading towards the principal metal binding site. Together with whole-cell copper quantitation and quantitative proteomics in a murine lung infection model, our data identify OprC as an abundant component of bacterial copper biology that may enable copper acquisition under a wide range of conditions.

## Introduction

Metals fulfil cellular functions that cannot be met by organic molecules and are indispensable for the biochemistry of life in all organisms. Copper is the third-most abundant transition metal in biological systems after iron and zinc. It has key roles as structural component of proteins or catalytic cofactor for enzymes [1], most notably associated with the biology of oxygen and in electron transfer. On the other hand, an excess of copper can be deleterious due to its ability to catalyse production of hydroxyl radicals [2,3]. Excessive copper may also disrupt protein structure by interaction with the polypeptide backbone, or via replacement of native metal cofactors from proteins, thus abolishing enzymatic activities via mismetallation [1,4,5]. Thus, cellular copper levels and availability must be tightly controlled. Bacterial copper homeostasis systems are well characterised [6]. Specific protein machineries are involved in fine-tuning the

deposited in the Protein Data Bank (http://www.ebi.ac.uk/pdbe/) with accession codes 6FOK (OprCWT), 6FOM (OprCAA), 6Z8Q (OprCWT Ag 8000 ev), 6Z91 (OprCC143A Ag 8800 eV), 6Z99 (OprCC143A Ag 9175 eV), 6Z8Y (OprCC143A Cu 8800 eV), 6Z8Z (OprCC143A Cu 9175 eV), 6Z8T (OprCH323A Ag 8800 eV), 6Z8U (OprCH323A Ag 9175 eV), 6Z8R (OprCM147H), 6Z8S (OprCM325H), 6Z9N (OprCH323A Cu 9175 eV), 6Z9Y (OprCH323A Cu 8800 eV).

**Funding:** SPB is supported by a Biotechnology and Biological Sciences Research Council (BBSRC, UK) grant (BB/R004366/1 to BvdB). TY acknowledges support from the Royal Commission for the Exhibition of 1851. The research leading to these results was in part conducted as part of the Translocation consortium (www.translocation.eu) and has received support from the Innovative Medicines Initiatives Joint Undertaking under Grant Agreement No. 115525, resources that are composed of financial contributions from the European Union's seventh framework programme (FP7/2007–2013) and European Federation of Pharmaceutical Industries and Associations companies in-kind contribution. BvdB would also like to acknowledge the Royal Society for salary support. The funders had no role in study design, data collection and analysis, decision to publish, or preparation of the manuscript.

**Competing interests:** The authors have declared that no competing interests exist.

**Abbreviations:** BCS, bathocuproinedisulfonic acid; DDM, n-dodecyl-beta-D-maltopyranoside; EDTA, ethylenediamine tetra-acetic acid; EPR, electron paramagnetic resonance; HCD, high-collision dissociation; ICP-MS, inductively coupled plasma mass spectrometry; IMAC, immobilised metal affinity chromatography; LDAO, lauryl-dimethylamine oxide; MFS, major facilitator superfamily; mgf, mascot generic file; MR, molecular replacement; OM, outer membrane; PRM, parallel reaction monitoring; SAD, single anomalous dispersion; SEC, size exclusion chromatography; SPF, specific pathogen-free; TBDT, TonB-dependent transporter; TEV, tobacco etch virus; WT, wild-type.

balance of intracellular copper trafficking, storage, and efflux according to cellular requirement, in such a way that copper is always bound to proteins. This control is executed by periplasmic and cytosolic metalloregulators, which activate transcription of periplasmic multi-copper oxidases, metallochaperones, copper-sequestering proteins [7,8], and transporters [9–11]. To date, relatively few families of integral membrane proteins have been validated as copper transporters, and these have different structures and transport mechanisms [12]. The $P_{1B}$-type ATPases such as CopA are responsible for Cu(I) efflux from the cytosol via several metal binding domains, using energy released from ATP hydrolysis [13–15]. A second class of copper export proteins are RND-type tripartite pumps such as CusABC, which efflux Cu(I) by utilising the proton-motive force [16–18]. Relatively, few copper influx proteins have been identified. The bacterial inner membrane copper importer CcoA is a major facilitator superfamily (MFS)-type transporter involved in fine-tuning the trafficking of copper into the cytosol and required for cytochrome c oxidase maturation [19,20]. The Ctr family of copper transporters is responsible for Cu(I) translocation into the cell without requiring external sources of energy [21]. However, Ctr homologues are found only in eukaryotes, and the molecular mechanisms by which copper ions enter gram-negative bacteria is largely unclear. The exception is copper import via metallophores like methanobactin, a small Cu-chelating molecule that is secreted by methanotropic bacteria and most likely taken up via TonB-dependent transporters (TBDTs), analogous to iron-siderophore [22].

*Pseudomonas aeruginosa* is a versatile and ubiquitous Gram-negative bacterium and a notorious opportunistic pathogen in humans that plays a major role in the development of chronic lung infection in cystic fibrosis patients [23,24]. *P. aeruginosa* has a number of TBDTs in the outer membrane (OM) dedicated to the acquisition of different iron-siderophore complexes such as pyochelin and pyoverdin [25]. In addition, *P. aeruginosa* contains another TBDT, termed OprC (*PA3790*), whose function has remained enigmatic. Nakae and colleagues suggested that OprC binds Cu(II) with micromolar affinities [26]. Transcription of OprC was found to be repressed in the presence of Cu(II) in the external medium under aerobic conditions [26–29], suggesting a role for OprC in copper acquisition. Very recently, the blue copper protein azurin was reported to be secreted by a *P. aeruginosa* Type VI secretion system and to interact with OprC, suggesting a role of the latter in Cu(II) uptake [29].

To shed light on the role of OprC in copper biology, we have determined X-ray crystal structures of wild-type (WT) and mutant OprC proteins in the absence and presence of copper and silver and characterised metal binding via inductively coupled plasma mass spectrometry (ICP-MS) and electron paramagnetic resonance (EPR). In addition, we have confirmed metal acquisition by OprC using whole-cell metal quantitation. OprC indeed has the typical structure of a TBDT, and differences between the Cu-loaded and Cu-free protein demonstrate changes in tertiary structure that likely lead to TonB interaction and copper import. Metal binding experiments and crystal structures of WT and mutant OprC proteins suggest that the unique metal binding site of OprC could enable import of both Cu(I) and Cu(II).

## Results

### OprC is a TonB-dependent transporter that binds ionic copper

The structure of OprC, crystallised with an N-terminal His7 purification tag under aerobic conditions in the presence of 2 mM CuCl₂, was solved using single wavelength anomalous dispersion (Cu-SAD), using data to 2.0 Å resolution (Methods; S1 Table, S1 Fig). As indicated by the successful structure solution, OprC contains a single bound copper and shows the typical fold of a TBDT, with a large 22-stranded β-barrel occluded by an N-terminal approximately 15 kDa plug domain that, like in other TBDTs, completely occludes the lumen of the barrel

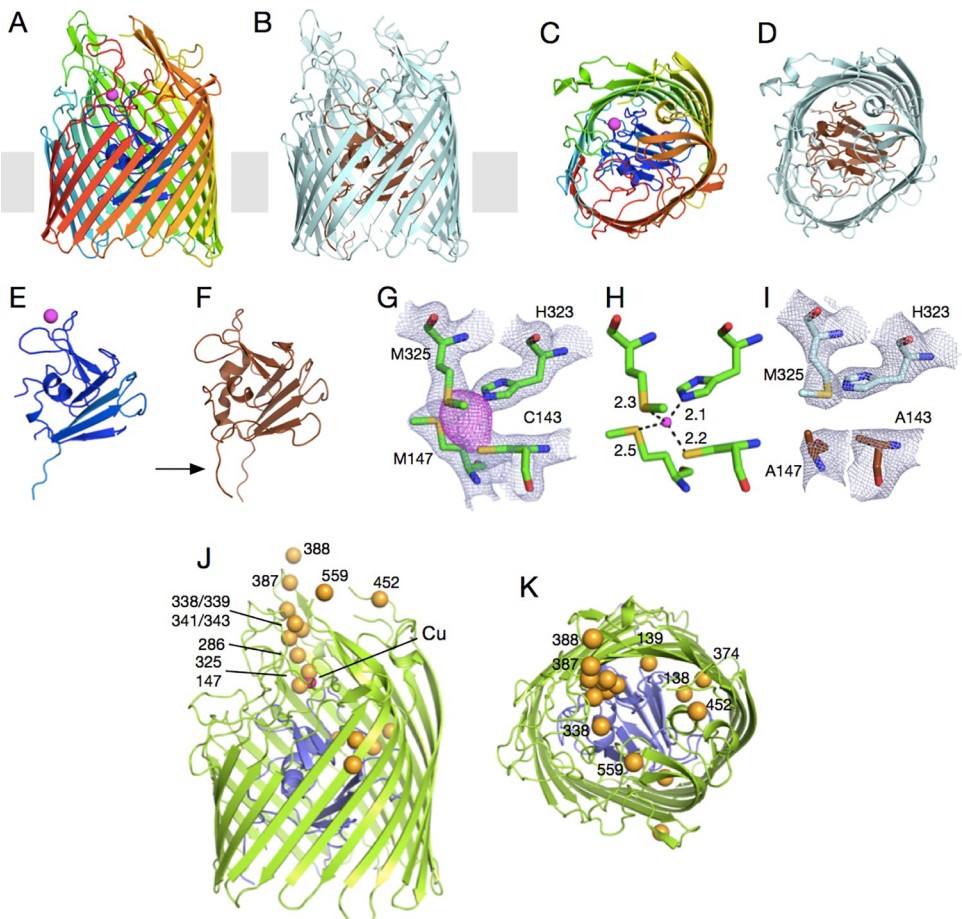

**Fig 1. OprC is a TBDT that binds ionic copper via an unprecedented CxxxM-HxM binding site.** Cartoon
representation of (A, C) Cu-loaded OprC and (B, D) Cu-free OprC (OprC$_{AA}$). (A, B) are viewed from the OM plane,
whereas the views for (C, D) are from the outside of the cell. The N-terminal plug domain is shown separately for both
forms (E, F). For Cu-OprC, structures are shown in rainbow from N-terminus (blue) to C-terminus (red); copper is
represented as a magenta sphere. Apo-OprC is coloured light cyan, with the plug rendered brown. The arrow in (F)
highlights the visibility of the Ton box in apo-OprC. (G) Stick models of copper-coordinating residues Cys143,
Met147, Met325, and His323. Electron density in grey mesh (2Fo-Fc map contoured at 2.0σ, carve = 2.0) is shown for
the binding site residues C/M-H/M and the copper atom (anomalous difference map shown in magenta, contoured at
3.0σ, carve = 2.25). (H) Distances between coordinating residues and metal show that copper is coordinated via 1
thiolate (from Cys), 2 thioethers (from Met), and 1 imidazole nitrogen from His. (I) Mutation of binding site residues
Cys143 and Met147 to alanines in OprC$_{AA}$ abolishes copper binding (2Fo-Fc map contoured at 2.0σ, carve = 2.0). (J,
K) OM plane (J) and extracellular views (K) showing the thioether atoms of all methionine residues in Cu-OprC as
yellow spheres. The copper atom, only visible in (J), is shown as a magenta sphere. OM, outer membrane; TBDT,
TonB-dependent transporter.

(Fig 1A, 1C, and 1E). The copper binding site comprises residues Cys143 and Met147 in the
plug domain and His323 and Met325 in the barrel wall. The CxxxM-HxM configuration,
which coordinates the copper in a tetrahedral manner (Fig 1G and 1H), is highly unusual and
has, to our knowledge, not been observed before in copper homeostasis proteins. A similar site
is present for one of the copper ions of the valence-delocalised Cu$_A$ dimer in cytochrome c oxi-
dase, where the copper ion is coordinated by 2Cys+1Met+1His [30,31]. Other similar sites are
class I Type I copper proteins like pseudoazurin and plastocyanin, where copper is coordinated
by 2His+1Cys+1Met [32]. Interestingly, and unlike class I Type I copper proteins, concen-
trated solutions and OprC crystals are colourless in the presence of Cu(II). Another notable

feature of the OprC structure becomes apparent when analysing the positions of the methionine residues. As shown in Fig 1J and 1K, out of the 16 visible methionines in OprC, 11 are organised in such a way that they form a distinct "track" leading from the extracellular surface towards the copper binding site. An additional methionine (Met558) is not visible due to loop disorder, but, given their positions, they will be a part of the methionine track. Considering that Cu(II) prefers nitrogen and oxygen as ligands while Cu(I) prefers sulphur, we propose that the methionine track might bind Cu(I) with low affinity and may guide the metal towards the principal binding site, which is at the bottom of the track (Fig 1J). Importantly, the anomalous difference maps of OprC crystallised with Cu(II) do not show any evidence for weaker, secondary copper sites (S1 Fig), demonstrating that the methionine track does not bind Cu(II).

## Copper binding by OprC is highly specific and near-irreversible

Following structure determination of copper-bound OprC, several attempts were made to produce a structure of copper-free OprC. First, the protein was purified and crystallised without added copper; however, this gave a structure that was identical to the one already obtained and contained bound copper that presumably originated from the LB medium. As expression in rich media always yielded OprC with approximately 45% to 80% copper as judged by ICP-MS, various attempts to lower the copper content were made. Removal of bound copper from purified protein with combinations of denaturants (up to 4.0 M urea) and ethylenediamine tetraacetic acid (EDTA) were not successful. Expression in minimal medium also yielded copper contents of approximately 45% to 60% (Fig 2A and 2B), but with much lower protein yields compared to rich medium.

Aerobic incubation of purified WT OprC (OprC$_{WT}$) with 45% copper occupancy in the presence of either 3 or 10 equivalents Cu(II) followed by size exclusion chromatography (SEC) demonstrate coelution of 1 equivalent copper (Fig 2B). Thus, the His7 tag does not bind Cu(II) with high affinity. Coincubation with 0.5 mM EDTA (approximately 50-fold excess) does not result in copper loading, suggesting that EDTA effectively withholds Cu(II) from OprC (S2 Fig). As-purified OprC does not contain zinc, the most common contaminant in metal-binding proteins, nor does it contain appreciable amounts of any other metals that could have been introduced during purification such as Ni and Fe, indicating that OprC is highly specific for copper (Fig 2A, S2 Fig). Indeed, incubation of purified OprC in the presence of 3 or 10 equivalents Zn does not result in zinc coelution (S2 Fig). To obtain copper-free OprC after purification from rich media, we constructed a variant (OprC$_{AA}$) in which the binding site residues Cys143 and Met147 were both mutated to alanines. Even after equilibration of OprC$_{AA}$ for 30 min with 3 or 10 equivalents Cu(II), no coelution with metal is observed (Fig 2B), indicating that high-affinity copper binding is completely abolished and confirming that the His7 tag does not bind Cu(II) with an affinity high enough to survive SEC, possibly due to the presence of 0.5 mM EDTA in the SEC buffer.

The fact that it is not possible to obtain copper-free WT protein, even after taking extensive precautions, suggests that copper binds to OprC with very high affinity. To explore this further, we performed a copper extraction assay with a large excess of bathocuproinedisulfonic acid (BCS) under reducing conditions (Methods). For copper-loaded OprC$_{WT}$, only 20% copper was removed after 24 h at room temperature, and the temperature had to be increased to 60°C to obtain near-quantitative extraction of copper (approximately 90% after 24 h) (Fig 2C). For reasons that are unclear, the orange-coloured $[Cu(BCS)_2]^{-3}$ complex was hard to separate from OprC, and BCS-treated OprC did not bind copper anymore, suggesting an irreversible change in the protein due to the harsh incubation conditions. Nevertheless, these results demonstrate that copper is kinetically trapped inside OprC and is, for all intents and purposes,

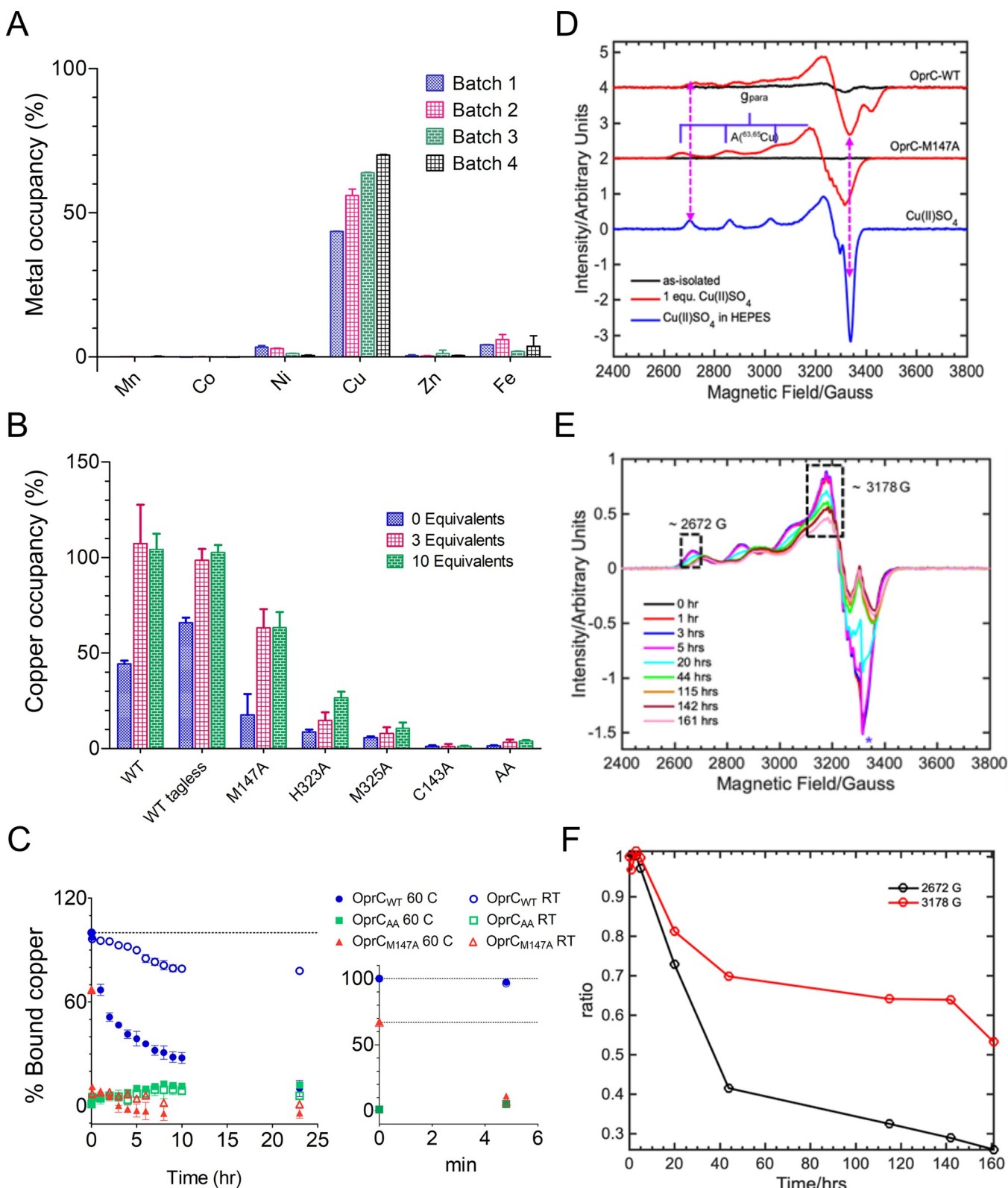

**Fig 2. OprC binds 1 equivalent copper near-irreversibly.** (A) Metal occupancy of LB-purified WT OprC by ICP-MS shows specific binding only to copper. Each colour indicates individual batches of protein purified from rich media. (B) Copper content of WT OprC and binding site mutant proteins before (blue) and after aerobic incubation with either 3 (pink) or 10 (green) equivalents Cu(II) for 30 min followed by analytical SEC. All proteins contain a N-terminal His7

tag except where stated. (C) Copper is kinetically trapped in OprC. Time course of copper extraction experiments showing % bound copper for OprC$_{WT}$ (blue), OprC$_{AA}$ (green), and OprC M147A (red), at RT (open symbols) and 60°C (filled symbols). The inset shows % bound copper in the first few minutes after starting the experiment. OprC$_{AA}$ served as a control. Dotted lines indicate initial occupancies of OprC$_{WT}$ and M147A. (D) Comparison of the cw-EPR spectra of OprC$_{WT}$ and OprC$_{M147A}$ mutant before (black traces) and after (red traces) addition of 1 equivalent Cu(II) solution. The blue trace shows the EPR spectrum of the Cu(II)SO$_4$ in Hepes buffer. All EPR spectra have been background subtracted. The double-headed magenta dotted arrows show the difference in the observed **g** and **A** tensor of OprC variants. The blue goal posts indicate the $^{63,65}$Cu-hyperfine splitting along the parallel region. Note that the starting copper equivalencies for these proteins were 0.6 (OprC$_{WT}$) and 0.1 (OprC$_{M147A}$), respectively. (E) EPR time course for OprC$_{M147A}$ after addition of 1 equivalent Cu (II). (F) Relative intensities of EPR signals at approximately 2,672 G and 3,178 G (black dotted rectangular boxes in the top panel) plotted as a function of time. Values shown are averages from 3 independent time courses. Underlying data for this figure can be found in S1 Data. EPR, electron paramagnetic resonance; ICP-MS, inductively coupled plasma mass spectrometry; RT, room temperature; SEC, size exclusion chromatography; WT, wild-type.

irreversibly bound. This is fully compatible with the consensus transport mechanism of TBDTs, in which the interaction with TonB, occurring after substrate binding, is required to disrupt the binding site and release the ligand [33].

## Conformational changes upon copper binding

The OprC$_{AA}$ structure was solved by molecular replacement (MR) using Cu-bound OprC$_{WT}$ as the search model (Fig 1B, 1D, 1F, and 1I). The binding site residues of both structures occupy very similar positions, indicating that the introduced mutations abolish copper binding without generating gross changes in the binding site. Superposition of the structures (S3A Fig and S3B Fig) shows that for the remainder of the protein, structural changes upon copper binding are confined to the vicinity of the copper binding site, with parts far removed virtually unchanged (overall Cα RMSD approximately 1.0 Å). The largest change is observed for loop L11, which undergoes an inward-directed motion of approximately 8.0 Å upon copper binding (S3D, S3E, and S4 Figs). A similar inward-directed but smaller change occurs for loop L8. Some loop tips (e.g., L4, L5, L6) in OprC$_{AA}$ lack electron density for a limited number of residues, suggesting increased mobility. Overall, the conformational changes of the loops upon copper binding likely decrease the accessibility of the copper binding site. However, the main reason why the bound copper is inaccessible to solvent is that the binding site residues Met147 and Met325, together with Asn145, effectively form a lid on the copper ion in the WT protein. In the double mutant, copper becomes solvent accessible due to the absence of the Met147 side chain (S3D, S3E, and S4 Figs).

The consensus mechanism for TonB-dependent transport postulates that ligand binding on the extracellular side generates conformational changes that are propagated to the periplasmic side of the plug and increase the periplasmic accessibility of the Ton box for subsequent interaction with TonB [34]. In OprC$_{AA}$, N-terminal density is visible up to Leu66 (i.e., the first 10 residues of the mature protein are disordered) including the Ton box ($^{68}$PSVVTGV$^{75}$), which is tucked away against the plug domain and the barrel wall. In Cu-OprC, the density between Glu88 and Pro94 is poor and hard to interpret, and, more importantly, no density is observed before Pro79, including the Ton box (Fig 1E and 1F, S3C Fig). Thus, while we cannot say conclusively that the Ton box is accessible to TonB in Cu-OprC, the structures do show that changes occur in the Ton box upon substrate binding. Thus, the structures of OprC in the absence and presence of ligand are consistent with the consensus TBDT mechanism. The observed position of the Ton box in OprC$_{AA}$, likely hard to reach from the periplasmic space, would prevent nonproductive interactions of TonB with transporters that do not have substrate bound [34].

## The OprC methionine track and the principal binding site bind Cu(I)

We next asked whether OprC also binds Cu(I). Since it is challenging to maintain copper in its +1 state during crystallisation, we used silver (Ag(I)) as a proxy for Cu(I) and determined the

co-crystal structure of WT OprC in the presence of 2 mM $AgNO_3$ (Methods). This is possible because the protein used for crystallisation only had approximately 60% copper occupancy. Data were collected at 8,000 eV, at which energy the anomalous signal of copper is very small (0.6 $e^-$, compared to 4.2 $e^-$ for Ag) (S1 Fig), the anomalous map of OprC WT crystallised in the presence of silver shows not one but 3 anomalous peaks. The first, very strong peak (Ag1; 23σ) is located at the same site as in OprC crystallised with Cu(II) and is coordinated by the same residues (Cys143, Met147, His323, and Met325; Fig 3A). The other 2 silver sites have lower occupancies (Ag2, approximately 10σ and Ag3, approximately 10σ) and are each coordinated by 3 methionines of the methionine track (Met286, Met339, and Met343 for Ag2; Met341, Met343, and Met348 for Ag3), as seen also, for example, for Cu(I) in the Ctr1 copper transporter [21]. Only 5 Met residues form the 2 sites (Met343 is shared between the two), and this, combined with the fact that the track has 11 or 12 methionines, suggests that there are more than 2 metal binding sites. The distance between Ag2 and Ag3 is approximately 6.5 Å, i.e., large enough for 2 metal ions to be bound simultaneously, consistent with the fact that Ag was present at approximately 10-fold excess during crystallisation (approximately 0.2 mM OprC and 2 mM Ag). Under physiological conditions, however, it seems more likely that only one Ag atom is bound to the track at any one time. We speculate that the sites closer to the irreversible binding site (Ag1) may have a higher affinity, driving metal towards this site. While direct measurement of the affinities of the methionine binding track sites would be challenging, the structural data suggest that the methionine track provides at least 2, and possibly more, binding sites for Ag(I), and, by extension, for Cu(I). Moreover, while the methionine track only binds Cu(I), the high-affinity CxxxM-HxM site likely binds both copper redox states.

Intriguingly, 3 out of the 5 "non-track" methionine residues (Met138 and Met139 in the plug, Met374 in the barrel wall) are located close together and on the same side of the barrel as the methionine track and the principal binding site (Fig 1K). Given that the nature of the TonB-dependent remodelling of the plug is not known, it is unclear whether a potential binding site formed by these residues could be part of the path taken by the metal towards the periplasmic space.

To obtain more information on the individual residue contribution to copper binding, we next generated the complete set of single alanine mutants of the principal binding site residues (C143A, M147A, H323A, and M325A) and determined copper binding via analytical SEC and ICP-MS. For all single mutants, the copper content after protein purification from LB-grown cells was below 10%, except for M147A (approximately 20%) (Fig 2B). Upon incubation with 3 or 10 equivalents Cu(II), various occupancies were obtained. C143A has no bound copper even after incubation with 10 equivalents Cu(II), suggesting that this residue has a crucial role. H323A (approximately 30%) and, in particular, M147A (approximately 60%), have relatively high occupancies after copper incubation and SEC, indicating that these residues contribute less towards binding. Of the 4 ligands, the M147 thioether is the furthest away from the copper in the crystal structure (Fig 1H), which may explain why it contributes the least to ligand binding. Interestingly, removal of bound copper is much faster in the M147A mutant compared to $OprC_{WT}$ (Fig 2C), suggesting that solvent exclusion by the M147 side chain (S3D Fig) is the main reason why copper is kinetically trapped in $OprC_{WT}$.

To shed additional light on the redox state of the bound copper, continuous wave EPR (cw-EPR) spectra were recorded on $OprC_{WT}$. Surprisingly, as-purified $OprC_{WT}$ containing approximately 0.6 equivalents copper was EPR silent (Fig 2D), demonstrating that the copper species present is Cu(I). The as-purified M147A protein, with approximately 0.1 equivalent copper, was EPR silent as well. We next loaded the M147A mutant with $CuSO_4$ to 1 equivalent, and EPR spectra were recorded over time. The observed EPR signal is different from the

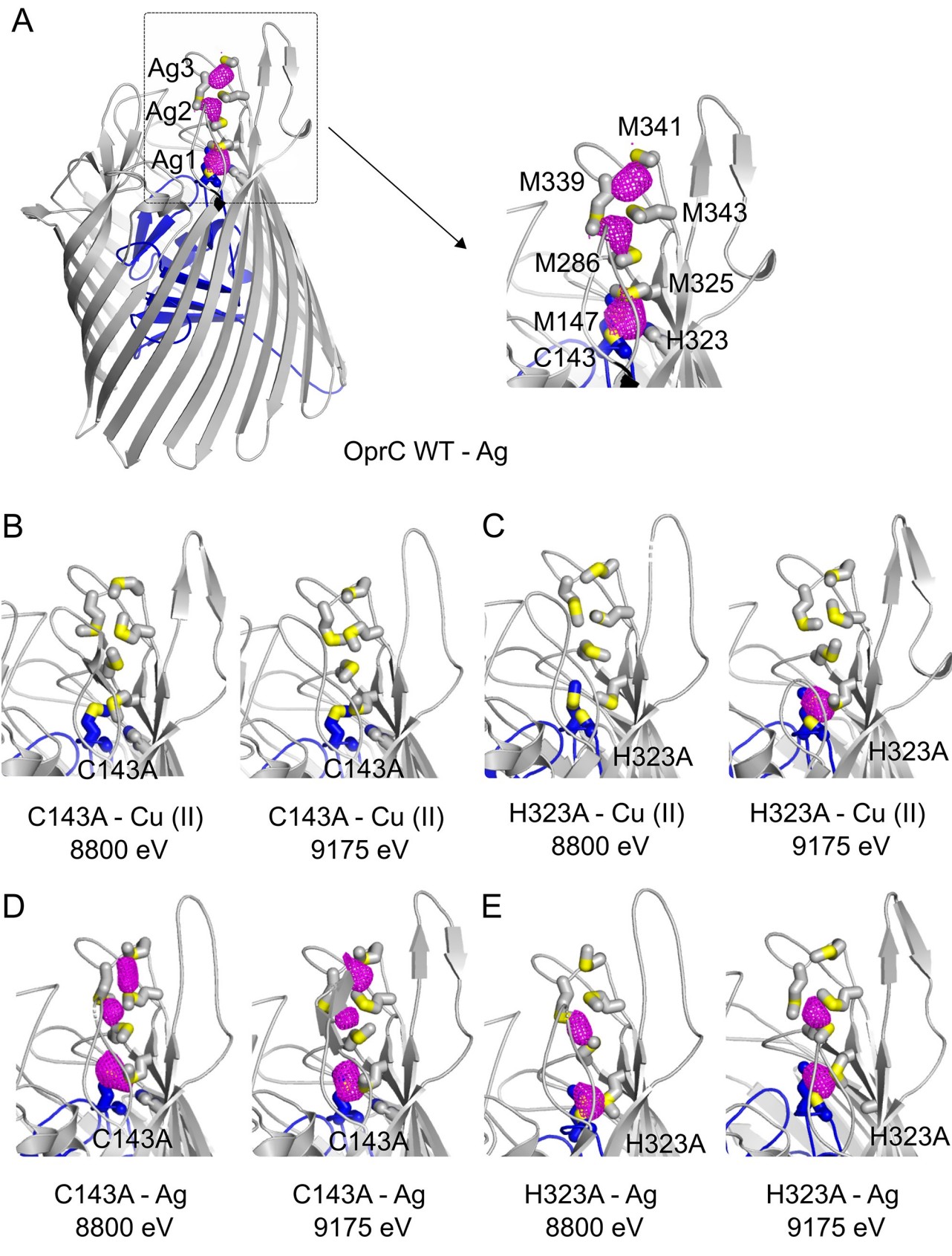

**Fig 3. OprC binds Ag at the high-affinity binding site and the methionine track.** (A-E) Anomalous difference maps of (A) OprC$_{WT}$, (B, D) C143A, and (C, E) H323A variants crystallised in the presence of (A, D, E) Ag or (B, C) Cu(II) and collected at different energies. The inset to (A) shows a close-up of the anomalous difference peaks (magenta) near the principal binding site in OprC$_{WT}$, with binding residues labelled and represented as stick models. Sulphurs are coloured yellow. For clarity, the metal used in cocrystallisation and the energy used for data collection are shown underneath each panel. The OprC plug domain is coloured blue. WT, wild-type.

standard CuSO$_4$ Cu(II) EPR signals, confirming that Cu(II) binds to the protein. The EPR spectra of the OprC-WT and OprC-M147A mutant show nicely resolved $^{63,65}$Cu(II) hyperfine coupling along the parallel region, due to the interaction of an unpaired electron spin ($S$ = ½) of Cu(II) with the nuclear spin of ($I$ = 3/2) of $^{63,65}$Cu nuclei, as indicated by the blue goal post in Fig 2D. Interestingly, the EPR signals decrease slowly upon prolonged incubation, suggesting that added Cu(II) is very slowly reduced to Cu(I) (Fig 2E and 2F). This, together with the possibility that OprC binds Cu(I) directly from the LB media, could be an explanation for the observation that as-purified OprC, expressed under aerobic conditions, contains reduced copper. However, it is clear that the observed reduction of Cu(II) is too slow to be physiologically relevant, obviating the need to find a mechanistic explanation.

## Cysteine is essential for high-affinity copper Cu(II) binding

While OprC$_{WT}$ and most single alanine mutants can be (partly) loaded via Cu(II) incubation, this is not the case for the C143A mutant (Fig 2B). We hypothesised that removal of the cysteine could lead to much lower affinity for Cu(II), so that after SEC, nothing remains bound. To provide support for this, we determined the crystal structures of the OprC C143A mutant cocrystallised with Cu(II) or silver Ag(I). For each crystal, datasets were collected at 8,800 eV and 9,175 eV to distinguish between both metals. Bound copper is expected to give a strong anomalous peak only at 9,175 eV (which is above the copper K edge at 8,979 eV), while bound silver will give comparable peaks at both energies (the silver L-III edge is at 3,351 eV). For C143A cocrystallised with Cu(II), no anomalous peaks are visible at both energies (Fig 3B), showing that Cu(II) binding is indeed abolished. Crucially, in the presence of silver, the same 3 anomalous peaks are visible as for WT OprC (compare Fig 3A and 3D), strongly suggesting that the C143A mutant can still bind Cu(I). Since Cu(II) prefers histidine nitrogen as ligands and Cu(II) binding sites often contain one or more His residues, we also cocrystallised the H323A mutant with Cu(II) and Ag(I). As shown in Fig 3C and 3E, one strong anomalous peak, at the high-affinity binding site, is observed with Cu(II), supporting the SEC data that the histidine is not required for Cu(II) binding. With Ag(I), 2 clear anomalous peaks are observed, suggesting that the H323A mutant can still bind Cu(I) at the principal site and at the methionine track.

## OprC mediates copper acquisition in *P. aeruginosa*

To demonstrate that OprC imports copper, we performed anaerobic growth experiments in *P. aeruginosa* with added copper. Given that *oprC* expression is repressed with excess external Cu(II) [26–29], we employed arabinose-inducible overexpression of His-tagged *oprC* via the broad range pHERD30 plasmid [35]. We complemented the PA14 Δ*oprC* strain with OprC$_{WT}$- and OprC$_{AA}$-containing plasmids and performed growth assays in rich media with empty vector as control. S5 Fig shows clear toxicity when OprC$_{WT}$ is overexpressed, even without Cu(II) addition. Surprisingly, expression of OprC$_{AA}$ was equally toxic as OprC$_{WT}$ overexpression, which indicates that the toxicity phenotype is caused by overexpression of OprC per se and is likely not linked to OprC function.

Since copper toxicity assays failed, we decided to determine *P. aeruginosa* whole-cell metal contents using ICP-MS. We observed no differences in copper content between the WT PA14

and Δ*oprC* strains in rich media without added copper (S6 Fig), suggesting that OprC is not expressed under these conditions. By contrast, cells expressing OprC$_{WT}$ from pHERD30 have more associated copper when compared to the empty vector control, under both aerobic and anaerobic conditions (Fig 4A). However, as shown by the toxicity phenotypes presented above, this could also be due to increased leakiness of cells as a result of plasmid-based OMP expression, a possibility that was not taken into account in a recent study [29]. However, cells expressing the OprC$_{AA}$ inactive mutant have copper levels similar to those of the control. Moreover, OprC$_{WT}$ and OprC$_{AA}$ are present at similar levels in the OM (Fig 4B), demonstrating that the different amounts of copper associated with the cells are not due to differences in protein levels. In addition, no substantial differences were detected for other divalent metals, confirming the in vitro experiments that OprC is specific for copper. These data, together with the fact that the OprC structures exhibit all the hallmarks of a bona fide TBDT (Fig 1, S3 Fig), strongly suggest that OprC is a copper importer in *P. aeruginosa*. Given the relatively modest differences in observed copper contents, cellular fractionation studies to experimentally demonstrate OprC-mediated copper import versus binding will be challenging. In addition, it is not known to which of the 3 *P. aeruginosa* TonB proteins OprC couples, and how expression of this TonB$_{OprC}$ is regulated, i.e., it is possible that OprC is fully loaded without copper being imported. It also seems clear that *P. aeruginosa* can acquire copper in the absence of OprC, either via one of the many OM channels expressed in *P. aeruginosa* or perhaps via (a) porin-independent pathway(s) as recently shown for antibiotics [36].

## OprC is abundant in *P. aeruginosa* during infection

Recent studies have suggested that OprC is important for virulence in *P. aeruginosa* and *Acinetobacter baumannii* [37,38]. These papers, however, did not provide a measure of the abundance of OprC during infection in vivo. We therefore decided to determine the abundance of the highly similar OprC proteins (50% sequence identity; S7 Fig) in *P. aeruginosa* UCBPP-PA14 and *A. baumannii* ATCC 19606 in infected lung tissues via a sensitive targeted proteomics approach with parallel reaction monitoring (Methods). The results showed that in both mouse and rat pneumonia models, OprC was present at 1,000 to 10,000 molecules per *P. aeruginosa* cell, making it one of the 5 most abundant TBDTs. As a comparison, the most abundant TBDT, FpvA (involved in iron-siderophore uptake), had 8,000 to 33,000 molecules per cell. In *A. baumannii*, OprC was less abundant in mouse and rat pneumonia models (40 to 400 molecules per cell), while the most abundant TBDTs BfnH and BauA were present at 500 to 3,000 molecules per cell.

## Discussion

Our data suggest that the TBDT OprC binds both Cu(I) and Cu(II) very tightly at an unusual CxxxM-HxM binding site that becomes solvent excluded upon metal binding, kinetically trapping the metal and precluding determination of metal binding affinities. The copper site most similar to that in OprC occurs in class I Type I copper proteins like cytochrome c oxidase, pseudoazurin and plastocyanin, electron transfer proteins that can coordinate both Cu(I) and Cu(II) via 2Cys+1Met+1His site or 2His+1Cys+1Met sites. Interestingly, the active site His117 in azurin renders the copper atom solvent inaccessible [39,40], reminiscent to the likely role of Met147 in OprC. The superficial similarity of the OprC binding site to that of (pseudo)azurin prompted us to generate the M147H and M325H OprC mutants in an attempt to convert OprC into a unique, blue copper transport protein. However, in the presence of added Cu(II), both mutants remain colourless, and comparison of the crystal structures with those of pseudoazurin and plastocyanin show small but most likely important differences in the geometries

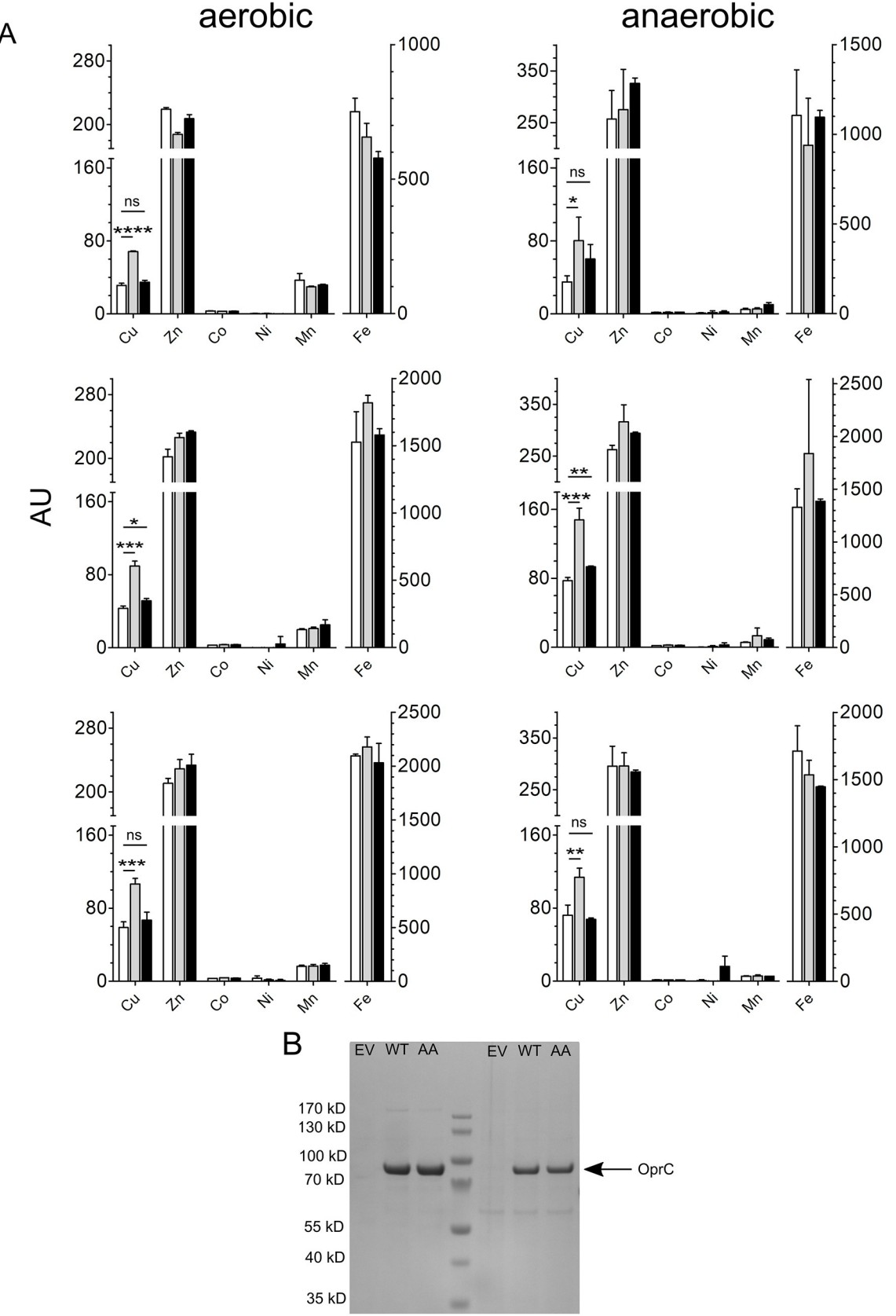

**Fig 4. Specific acquisition of copper by OprC.** (A) Whole-cell metal content of PA14 *ΔoprC* cells overexpressing empty vector (white bars) OprC$_{WT}$ (grey bars) and OprC$_{AA}$ proteins (black bars) analysed via ICP-MS. Cell associated metal content was determined in cells grown in rich media supplemented with 100 mM sodium nitrate (no added copper) under both aerobic (left panels) and anaerobic conditions (right panels). The 3 biological replicates are plotted separately due to differences in absolute metal levels. Reported values are averages ± SD ($n = 3$). Significant levels were analysed via unpaired 2-tailed $t$ test. ns, not significant ($p \geq 0.05$); *, $p \leq 0.05$; **, $p \leq 0.01$; ***, $p \leq 0.001$; ****, $p \leq 0.0001$. (B) SDS-PAGE gel of pHERD30-overexpressed OprC$_{WT}$ and OprC$_{AA}$ proteins in PA14 *ΔoprC* after IMAC. Masses from the molecular weight marker are shown on the left. Underlying data for this figure can be found in S1 Data. ICP-MS, inductively coupled plasma mass spectrometry; IMAC, immobilised metal affinity chromatography; ns, not significant; WT, wild-type.

of the active sites (S8 Fig). All 4 residues in the OprC binding site contribute to copper binding, but not to the same extent and, in some cases, not equally for both copper redox states. This is illustrated by Cys143, which is essential for high-affinity Cu(II) binding but dispensable for Cu(I). By contrast, removal of His323 still allows Cu(II) and Cu(I) binding. The presence of a methionine binding track, constituting several low-affinity binding sites for Ag(I)/Cu(I) but not for Cu(II), suggests that metal delivery to OprC may occur in different ways for Cu(I) and Cu(II). Methionines also coordinate Cu(I) in other copper transporters, such as the bacterial CusABC and CopA exporters and the eukaryotic Ctr1 copper importer [17,18,21,41,42]. These transporters have been shown to also transport silver, but structures with silver have been determined only for the Cus system [17,41]. By analogy, our structural data therefore strongly suggest that OprC can also import silver. The ability of OprC to take up Cu(I) could be important for biofilms, which are anaerobic to various degrees depending on the location inside the biofilm [43,44]. Oxygen tension also reduces as lung chronic disease mediated by *P. aeruginosa* progresses, turning airway mucus into an anaerobic environment in cystic fibrosis patients that will favour the availability of Cu(I) [45]. *P. aeruginosa* is capable of anaerobic respiration by using nitrate, nitrite, or nitrous oxide as terminal electron acceptor [46], and OprC has been shown to be induced under anaerobic conditions [47].

With respect to Cu(II), the affinity of 2.6 μM reported by an early study [26] most likely resulted from nonspecific binding, given that (i) copper is kinetically trapped and (ii) the rich media used [48] to culture *P. aeruginosa* would have generated OprC with high copper occupancy. A recent study in *P. aeruginosa* proposed a novel copper uptake mechanism in copper-limited conditions, which involves secretion of the copper binding protein azurin by a CueR-regulated Type VI secretion system. The secreted azurin would scavenge Cu(II) from the environment and load it onto OprC via a direct interaction, conferring a competitive advantage under copper-limiting conditions [29]. Delivery by azurin would be an efficient way to load OprC with Cu(II) and would presumably not require any low-affinity sites to guide the metal to the principal binding site as for Cu(I). However, the pulldown experiment done by Han and colleagues to show the azurin-OprC interaction was done with OprC folded in vitro from inclusion bodies [29]. Given that TBDTs are hard to fold in vitro due to their large size and complex architecture, and no attempts were made to assess the functionality of the obtained OprC, its interaction with azurin remains to be confirmed. In our hands, no complex formation between OM-purified, functional OprC and azurin was observed via SEC.

Given that copper and, in particular, the more toxic Cu(I), is a known antimicrobial, the presence of bacterial proteins dedicated to copper acquisition such as OprC might be problematic under certain conditions. Indeed, it is thought that, in contrast to iron that is withheld from a pathogen by the host during infection, elevated levels of host-derived copper in, e.g., macrophages, could be an alternative "nutritional immunity" antimicrobial response [49]. In this model, bacterial virulence would be attenuated by mutations, particularly in transporters, which cause copper sensitivity [49]. However, recent data suggest that deletion of *oprC* results in reduced quorum sensing, impaired motility, and lower virulence of *P. aeruginosa* in mice,

leading to the proposal that the presence of OprC is critical for virulence [37]. In addition, another recent study reported decreased virulence of an *A. baumannii* *oprC* knockout in mice [38]. The decreased virulence of *oprC* knockouts in these studies appears at odds with what one would expect from the copper nutritional immunity model [49], as is our proteomics data showing that OprC is very abundant in a *P. aeruginosa* mouse infection model. While we did not assess *oprC*-dependent virulence, these data do suggest that copper is withheld by the host under many conditions, requiring the presence of an abundant, highly specific TBDT. We speculate that, given that copper is bound near-irreversibly (Fig 2), OprC could provide a reservoir of bound copper on the cell surface, to be imported upon interaction with TonB$_{OprC}$ only when needed. While much still needs to be learned, it is clear that regulation of any copper import protein is crucial, possibly both at the gene and protein level. Unfortunately, and in contrast to the many copper stress genes that, as part of the CopR or CueR regulons, are up-regulated under aerobic conditions during copper stress [9,11,27,28,50,51], nothing is known about how *oprC* is down-regulated during such stress. Intriguingly, *oprC* (*PA3790*) is in an operon with *PA3789*, which encodes for an inner membrane protein that is orthologous to *P. aeruginosa* FoxB, which was very recently shown to be a reductase of the iron chelated by siderophores [52]. The FoxB active site is located in the periplasmic space, suggesting that PA3789 may be a reductase for OprC-imported Cu(II). Another protein strongly down-regulated during copper stress is PA5030, which is an MFS transporter with a large number of His+Met residues (26 out of 438 residues), suggesting that it could mediate copper delivery to the cytoplasm, possibly in concert with OprC and an as yet unidentified periplasmic protein [27].

OprC is the first example of a TBDT that mediates copper import without a metallophore. The TBDT with the closest substrate specificity to OprC is the zinc-specific ZnuD from *Neisseria meningitidis*, the structure of which has been solved but for which no transport data are available [53]. Large structural differences between OprC and ZnuD exist for the extracellular loops (overall Cα RMSD approximately 5.9 Å). ZnuD has several discrete low-affinity binding sites that may guide the metal towards the high-affinity binding site [53]. In OprC, a distinctive "methionine track" provides low-affinity binding sites to guide copper to the high-affinity site. Interestingly, while the extracellular loops between OprC and ZnuD are very different and the overall sequence identity is only 28%, the metal binding sites are located at very similar positions and only 2.8 Å apart (S9 Fig), suggesting that the transport channel formed via TonB interaction may be similar. Inspection of the ZnuD structure shows that the zinc binding site is excluded from solvent, and we propose that the zinc ion in ZnuD is kinetically trapped, analogous to copper in OprC.

OprC shares approximately 60% identity to NosA from *Pseudomonas stutzeri*, for which no structure is available. Like OprC, NosA is expressed under anaerobic conditions and repressed in the presence of μM concentrations Cu(II) [54–56]. *P. stutzeri* NosA antibodies did not react with *P. aeruginosa* [54], but our structure identifies NosA as an OprC ortholog, since the CxxxM-HxM copper binding motif and some of the methionine track residues are conserved (S7 Fig). NosA is important during denitrification in *P. stutzeri* JM300 and was proposed to load copper either directly or indirectly to the periplasmic N$_2$O reductase NosZ [54–56]. However, a more recent report for a different *P. stutzeri* strain found no difference between NosZ activity and copper content for a *nosA* knockout [57]. In addition, OprC/NosA also occurs in a number of nondenitrifying Proteobacteria such as *Salmonella enterica*, *Klebsiella pneumoniae*, and *A. baumannii* (S7 Fig), showing that NosZ maturation is not a general function of OprC. The occurrence of OprC in some (e.g., *S. enterica*) but not in other (e.g., *Escherichia coli*) Enterobacteria is intriguing, given that the OM of all Enterobacteria is relatively permeable to small polar molecules due to abundant general porins such as OmpC [58].

## Methods

### Recombinant production of *Pseudomonas aeruginosa* OprC

The mature version of the gene coding for *oprC* of *P. aeruginosa* PAO1 (UniProt ID; PA3790) [59], starting with His56 as determined by Nakae and colleagues [26], was synthesised to include a 7 × His tag at the N-terminus (Eurofins, United Kingdom), cloned into the pB22 arabinose-inducible expression vector [60] and transformed into chemically competent *E. coli* DH5α cells. After expression and processing by signal peptidase, the N-terminal sequence of this construct is NVRLQHHHHHHHLEAEEHSQHQ-. A second version of this construct was constructed in a pB22 version containing a tobacco etch virus (TEV) site after the His7--tag. Correct sequences were confirmed by DNA sequencing (Eurofins, UK) using both forward and reverse plasmid-specific primers. The OprC$_{AA}$ mutant was produced by changing the key amino acids Cys143 and Met147 to alanine residues using the KLD Quickchange site-directed mutagenesis kit (New England Biolabs, UK) and specific primers containing both mutation sites (forward: 5′-tcgcgcggatgcaccaaccagctatattagc-3′; reverse: 5′-ttcggggcggcgccaag-catcatgc-3′). The single mutants C143A, M147A, H323A, M325A, M147H, and M325H were made in similar ways.

OprC recombinant protein production and purification was performed as follows: *E. coli* C43 Δ*cyo* was electroporated with expression vector, recovered for 60 min in LB (Sigma, UK) at 37°C, and plated on LB agar (Sigma, UK) containing 100 μg mL$^{-1}$ ampicillin (Melford, UK). Transformants were cultured in LB medium or in LeMasters-Richards (LR) minimal medium with glycerol (2–3 g/l) as carbon source. All media contained 100 μg mL$^{-1}$ ampicillin. For rich media, cells were grown (37°C, 180 rpm) until OD600 approximately 0.6, when protein expression was induced with 0.1% arabinose for 4 to 5 h at 30°C or overnight at 16°C (150 rpm). For LR media, a small overnight preculture in LB was used at 1/100 v/v to inoculate an LR-medium preculture early in the morning (typically 1 ml preculture for 100 ml of cells was used), which was grown during the day at 37°C. After late afternoon inoculation, large-scale cultures (typically 6 to 8 l) were grown overnight at 30°C until OD 0.4 to 0.7, followed by induction with 0.1% arabinose at 30°C for 6 to 8 h. Cells were harvested by centrifugation (5,000 rpm, 20 min, 4°C), and pellets homogenised in 20 mM Tris (Sigma), 300 mM NaCl (Fisher) (pH 8.00) (TBS buffer), in the presence of 10 mM EDTA (Sigma). Cells were broken by one pass through a cell disruptor (Constant Systems 0.75 kW operated at 23 kpsi), centrifuged at 42,000 rpm for 45 min at 4°C (45Ti rotor; Beckman), and the resulting total membrane fraction was homogenised in TBS buffer containing 1.5% lauryl-dimethylamine oxide (LDAO) (Sigma, UK). Membrane proteins were extracted by stirring (60 min, 4°C), centrifuged (42,000 rpm in 45Ti rotor, 30 min, 4°C), and the membrane extract was loaded on a Chelating Sepharose Fast Flow bed resin (approximately 10 ml; GE Healthcare, UK) previously activated with 200 mM NiCl$_2$ (Sigma) and equilibrated in TBS containing 0.15% n-dodecyl-beta-D-maltopyranoside (DDM). After washing with 15 column volumes buffer with 30 mM imidazole, protein was eluted with 0.25 M imidazole buffer (Fisher), incubated with 20 mM EDTA (30 min, 4°C), and loaded on a Superdex 200 16/600 size exclusion column equilibrated with 10 mM HEPES, 100 mM NaCl, 0.05% DDM, 10 mM EDTA (pH 7.5). Peak fractions were pooled and concentrated using a 50 MWCO Amicon filter (Millipore, UK), analysed on SDS-PAGE, flash-frozen in liquid nitrogen, and stored at −80°C. Typical yields of purified WT and most mutant OprC proteins ranged between 2 and 5 mg per l media grown at 16°C. All media and buffer components were made in fresh milli-Q water.

Protein preparations intended for crystallisation trials were pooled and buffer-exchanged to 10 mM HEPES, 100 mM NaCl, 0.4% tetraethylene glycol mono-octyl ether (C$_8$E$_4$) (Anatrace, United States) (pH 7.5). NaNO$_3$ was substituted for NaCl for protein preparations intended

for crystal trials with silver in order to avoid formation of insoluble AgCl. Protein preparations to be used for metal analysis after removal of the His-tag underwent a slightly different protocol. The elution fraction from immobilised metal affinity chromatography (IMAC) was buffer-exchanged to 50 mM Tris, 0.5 mM EDTA, 0.2 mM TCEP, 100 mM NaCl, 0.05% DDM (Anatrace, US) (pH 7.50) and submitted to TEV protease digestion (ratio 1 mg TEV: 10 mg protein, 4°C, overnight). Samples were submitted to a second IMAC column, where flow-through and wash fractions were combined for the subsequent SEC step in 10 mM HEPES, 100 mM NaCl, 0.05% DDM, 0.5 mM EDTA (pH 7.5). Protein concentration was determined by BCA assay (Thermo Scientific, UK) and by UV/Vis absorbance at 280 nm (considering OprC $E_{0.1\%}$ = 1.6 as determined by ProtParam).

## In vitro metal binding assays and inductively coupled plasma mass spectrometry (ICP-MS)

OprC samples intended for metal binding assays were exchanged into respective chelex-treated buffers without EDTA and were equilibrated with different equivalents of Cu(II) or Zn, for 30 min at room temperature ($n$ = 3). Protein concentrations used were in the range of 10 to 20 μM. Samples were loaded on an analytical Superdex 200 Increase 10/300GL (GE Healthcare) column, equilibrated in 10 mM HEPES, 100 mM NaCl, 0.05% DDM, 0.5 mM EDTA (pH 7.5). Size exclusion peaks were pooled, concentrated, and quantified for protein by UV absorbance at 280 nm. Protein samples were diluted 10-fold in 2.5% $HNO_3$. Analytical metal standards of 0 to 500 ppb were prepared by serial dilution from individual metal stocks (VWR, UK) and were matrix-matched to protein samples. Samples and standard curves were analysed by ICP-MS using a Thermo X-series instrument (Thermo Fisher Scientific). OprC WT samples were screened for the presence of $^{65}$Cu, $^{55}$Mn, $^{59}$Co, $^{60}$Ni, $^{65}$Cu, $^{66}$Zn, and rest were typically screened for the presence of $^{65}$Cu and $^{66}$Zn. The increase in copper content in "as-purified" tagless WT protein (Fig 3B) is most likely due to the use of a second IMAC column after tag cleavage. In addition to the additional handling steps that could have increased copper content, the $NiCl_2$ used for the IMAC column might contain traces of copper.

## Copper extraction (demetallation) experiments

$OprC_{WT}$, $OprC_{AA}$, and M147A samples were incubated with 3 equivalents copper for 30 min and then loaded onto a Superdex S-200 Increase 10/300GL column equilibrated with 10 mM HEPES, 100 mM NaCl, 0.05% DDM, 0.5 mM EDTA (pH 7.5). Peak fractions were pooled, concentrated, and quantified by UV absorbance at 280 nm. Samples were exchanged into respective chelex-treated buffer without EDTA. For demetallation experiments, 15 to 20 μM of copper-bound proteins were taken in duplicates and incubated with 100-fold excess of the copper chelator BCS (Sigma) and 100-fold excess of the reducing agent hydroxyl amine ($NH_2OH$) (Sigma) at 60°C and room temperature. BCS is a high-affinity Cu(I) chelator ($\log\beta_2$ 20.8) and forms a 2:1 complex with Cu(I), namely $[Cu(BCS)_2]^{-3}$, with a molar extinction coefficient of 13,300 $cm^{-1}$ $M^{-1}$ at 483 nm, enabling quantitation of Cu(I) [61].

## Protein crystallisation, data collection, and structure determination

Sitting-drop crystallisation trials were set up using a Mosquito crystallisation robot (TTP Labtech) with commercial screens (MemGold1 and MemGold2, Molecular Dimensions) at 20°C. To obtain the initial structure of Cu-bound OprC, the protein (approximately 12 mg/ml) was incubated with 3 mM $CuCl_2$ for 1 h at room temperature, followed by setting up crystallisation trials. A number of initial hits were obtained and were subsequently optimised by manual hanging drop vapour diffusion using larger drops (typically 1 to 1.5 μl protein + 1 μl reservoir).

Well-diffracting crystals (approximately 3 Å resolution at a home source) were obtained in 0.1 M NaCl/0.15 M NH₄SO₄/0.1 M MES (pH 6.5)/18% to 22% PEG1000. Crystals were cryoprotected with mother liquor lacking $CuCl_2$ containing 10% PEG400 for approximately 5 to 10 s and flash-frozen in liquid nitrogen. Diffraction data were collected at Diamond Light Source (Didcot, UK) at beamline i02. For the best crystal, belonging to space group C222₁, 720 degrees of data were collected at an energy of 8,994 eV, corresponding to the K-edge of copper (S1 Table). Data were autoprocessed by xia2 [62]. The structure was solved via single anomalous dispersion (SAD) via AUTOSOL in Phenix [63]. Two copper sites were found, one for each OprC molecule in the asymmetric unit (S1 Fig). The phases were of sufficient quality to allow automated model building via Phenix AUTOBUILD, generating approximately 60% of the structure and using data to 2.0 Å. The remainder of the structure was built manually via iterative cycles of refinement in Phenix and model building in COOT [64]. Metal coordination was analysed by the Check-my-metal server [65]. The final refinement statistics are listed in S1 Table. Subsequently, crystals were also obtained without any copper supplementation of the protein. These were isomorphous to those described above and obtained under identical crystallisation conditions. MR indicated the presence of copper and an identical structure to that obtained above. OprC$_{AA}$ crystals (approximately 10 mg/ml protein) were obtained and optimised by hanging drop vapour diffusion as described above, and diffraction-quality crystals were obtained in the same conditions as for Cu-OprC, i.e., 0.1 M sodium chloride/0.15 M ammonium sulphate/0.01 M MES sodium (pH 6.5)/19% (w/v) PEG1000. Interestingly, however, the OprC$_{AA}$ crystals belong to a different space group (P22₁2₁), most likely as a result of the structural differences between both OprC variants. Diffraction data were collected at Diamond Light Source (Didcot, UK) at beamline I24. Diffraction data were processed in XDS [66]. The structure was solved by MR using Phaser, with WT OprC as the search model. Model building was done in COOT and refinement in Phenix. As for Cu-OprC, the data collection and refinement statistics are shown in S1 Table. C143A and H323A proteins (approximately 10 to 12 mg/ml protein) were incubated with 2 mM $CuSO_4$ at room temperature for 1 h, followed by cocrystallisation. Diffracting crystals for both C143A and H323A in the presence of copper were obtained in 0.34 M ammonium sulphate/0.1 M sodium citrate (pH 5.5)/12% to 16% w/v PEG 4000 and were cryoprotected using mother liquor lacking $CuSO_4$ and with 25% ethylene glycol for approximately 5 to 10 s and flash-frozen in liquid nitrogen. M147H and M325H crystals were obtained in the same condition as those for WT OprC. For cocrystallisation with silver, OprC proteins were incubated with 2 mM AgNO3 for 1 h at room temperature, followed by cocrystallisation. Well-diffracting OprC$_{WT}$ crystals with silver were obtained under the same conditions as in the presence of copper. For the best OprC$_{WT}$ crystal, belonging to space group C222₁, 999 degrees of data were collected at an energy of 8,000 eV to obtain anomalous signals for Ag. C143A and H323A crystals (approximately 10 to 12 mg/ml protein) with Ag were obtained from 0.2 M choline chloride/0.1 M Tris (pH 7.5)/12% to 16% w/v PEG 2000 MME and 0.5 M potassium chloride/0.05 M HEPES (pH 6.5)/12% to 16% v/v PEG 400, respectively. Crystals were cryoprotected for 5 to 20 s with mother liquor lacking $AgNO_3$ but containing 25% ethylene glycol for C143A and 20% PEG 400 for H323A. For C143A and H323A crystallised in the presence of copper or silver, datasets of 360 degrees each were collected at energies of 8,800 and 9,175 eV, using different parts of the same crystal (S2 and S3 Tables).

## Electron paramagnetic resonance spectroscopy

EPR measurements were carried out using a Bruker ELEXSYS-E500 X-band EPR spectrometer operating in continuous wave mode, equipped with an Oxford variable-temperature unit and

ESR900 cryostat with Super High-Q resonator. All EPR samples were prepared in quartz capillary tubes (outer diameter; 4.0 mm, inner diameter 3.0 mm) and frozen immediately in liquid $N_2$ until further analysis. The experimental setup and conditions were similar to those reported previously [67]. The low temperature EPR spectra were acquired using the following conditions: sweep time of 84 s, microwave power of 0.2 mW, time constant of 81 ms, average microwave frequency of 9.44 GHz, and modulation amplitude of 5 G, T = 20 K. The concentration of $OprC_{WT}$ and M147A varied from 210 to 260 μM in 10 mM HEPES, 100 mM NaCl, 0.03% DDM (pH 7.5).

## Determination of whole-cell metal content

Whole-cell metal content was determined as described previously. Briefly, overnight bacterial cultures of overexpressed $OprC_{WT}$ and $OprC_{AA}$ (with empty vector as control) in PA14 Δ*oprC* background were diluted with 1:100 fresh LB supplemented with 100 mM $NaNO_3$ and were grown to an OD of around 1.0 at 37˚C. Cultures (25 ml) were pelleted and were washed twice in TBS and once in 20 mM Tris, 0.5 M sorbitol, and 200 uM EDTA (pH 7.5). The cell pellets were digested in 1 ml of 68% conc. nitric acid. Digested sample pellets were diluted 10-fold in 2% nitric acid (prepared in chelex-treated milli-Q water) and were analysed by ICP-MS. Results were corrected for ODs and dilution factors. Protein levels in the OM were verified by IMAC. Briefly, 0.5 l of $OprC_{WT}$ and $OprC_{AA}$ overexpressing strains (with empty vector as control) in the *P. aeruginosa* Δ*oprC* background strain were grown in LB (supplemented with 100 mM $NaNO_3$ and 0.1% arabinose) for 6 h to $OD_{600}$ approximately 1.0, followed by cell harvesting, cell lysis, and purification as described above for *E. coli*.

## In vivo metal toxicity assays

For metal toxicity assays in *P. aeruginosa*, overexpressed OprC WT, C143A, and AA strains using broad-range plasmid pHERD30 (with empty vector as control) in PA14 Δ*oprC* were used and assays were performed in anaerobic conditions. The Cu(II) ($CuSO_4$) range tested varied from 0 to 7 mM. Cultures in triplicates were inoculated with 1:100 of the precultures grown in anaerobic conditions (LB with 100 mM sodium nitrate). Growth curves of final volume 200 μl were set up in 96-well Costar culture plate (Sigma Aldrich) and sealed inside an anaerobic chamber (Don Whitley Scientific, A35 workstation). Growth was monitored at 600 nm using an Epoch plate reader (Biotek Instruments Ltd) at 37˚C. Time points were collected with 30 min intervals and experiments were performed in triplicates.

## Animal infection models

Intratracheal instillation model: Specific pathogen-free (SPF) immunocompetent male Sprague–Dawley rats weighing 100 to 120 g or male CD-1 mice weighing 20 to 25 g were infected by depositing an agar bead containing around $10^7$ colony-forming units *A. baumannii* ATCC 19606 and *P. aeruginosa* UCBPP-PA14, deep into the lung via nonsurgical intratracheal intubation [68]. In brief, animals were anaesthetised with isoflurane (5%) and oxygen (1.5 l/min) utilising an anaesthesia machine. Depth of anaesthesia was evaluated by absence of gag reflex; if the reflex was present, the animal was placed back under anaesthesia until the reflex disappeared. No animals were utilised until they were fully anaesthetised. Animals were infected via intrabronchial instillation of molten agar suspension (rats- 100 μl) (mice- 20 μl) via intratracheal intubation and then allowed to recover. Animals were returned in their home cages and observed until recovered from anaesthesia. At 24-h postinfection, animals were killed, and lung was homogenised in sterile saline using a lab blender. All procedures are in accordance with protocols approved by the GSK Institutional Animal Care and Use Committee (IACUC)

and meet or exceed the standards of the American Association for the Accreditation of Laboratory Animal Care (AAALAC), the US Department of Health and Human Services, and all local and federal animal welfare laws.

## Sample workup for proteomics

The sample workup protocol was optimised to deplete host material while maintaining *A. baumannii* and *P. aeruginosa* viability until lysis. All buffers and equipment were used at 0 to 4˚C to minimise proteome changes during sample workup. The sample volume (maximum of 1 ml) was estimated, and an equal volume of 1% Tergitol in PBS was added followed by vigorous vortexing for 30 s. After centrifugation at $500 \times g$ for 5 min, the supernatant was transferred to a fresh tube, and the pellet was extracted again with 2 ml 0.5% Tergitol in PBS. The supernatant was combined with the first supernatant and centrifuged at $18,000 \times g$ for 5 min. The pellet was washed with 2 ml and again centrifuged at $18,000 \times g$ for 5 min. The supernatant was removed, and the pellet was resuspended in 100 μl 5% sodium deoxycholate, 5 mM Tris (2-carboxyethyl) phosphine hydrochloride, 100 mM $NH_4HCO_3$. The sample was incubated at 90˚C for 1 min and then stored at −80˚C. Samples were thawed and sonicated for $2 \times 20$ s (1-s interval, 100% power). Proteins were alkylated with 10 mM iodoacetamide for 30 min in the dark at room temperature. Samples were diluted with 0.1 M ammonium bicarbonate solution to a final concentration of 1% sodium deoxycholate before digestion with trypsin (Promega) at 37˚C overnight (protein to trypsin ratio: 50:1). After digestion, the samples were supplemented with TFA to a final concentration of 0.5% and HCl to a final concentration of 50 mM. Precipitated sodium deoxycholate was removed by centrifugation at 4˚C and 14,000 rpm for 15 min. Peptides in the supernatant were desalted on C18 reversed phase spin columns according to the manufacturer's instructions (Macrospin, Harvard Apparatus), dried under vacuum, and stored at −80˚C until further processing.

## Parallel reaction monitoring

Heavy proteotypic peptides (JPT Peptide Technologies GmbH) were chemically synthesised for *A. baumannii* and *P. aeruginosa* OM proteins. Peptides were chosen dependent on their highest detection probability and their length ranged between 7 and 20 amino acids. Heavy proteotypic peptides were spiked into each sample as reference peptides at a concentration of 20 fmol of heavy reference peptides per 1 μg of total endogenous protein mass. For spectrum library generation, we generated parallel reaction monitoring (PRM) [69] assays from a mixture containing 500 fmol of each reference peptide. The setup of the μRPLC-MS system was as described previously [70]. Chromatographic separation of peptides was carried out using an EASY nano-LC 1000 system (Thermo Fisher Scientific) equipped with a heated RP-HPLC column (75 μm × 37 cm) packed in-house with 1.9 μm C18 resin (Reprosil-AQ Pur, Dr. Maisch). Peptides were separated using a linear gradient ranging from 97% solvent A (0.15% formic acid, 2% acetonitrile) and 3% solvent B (98% acetonitrile, 2% water, 0.15% formic acid) to 30% solvent B over 60 min at a flow rate of 200 nl/min. Mass spectrometry analysis was performed on Q-Exactive HF mass spectrometer equipped with a nanoelectrospray ion source (both Thermo Fisher Scientific). Each MS1 scan was followed by high-collision dissociation (HCD) of the 10 most abundant precursor ions with dynamic exclusion for 20 s. Total cycle time was approximately 1 s. For MS1, 3e6 ions were accumulated in the Orbitrap cell over a maximum time of 100 ms and scanned at a resolution of 120,000 FWHM (at 200 m/z). MS2 scans were acquired at a target setting of 1e5 ions, accumulation time of 50 ms, and a resolution of 30,000 FWHM (at 200 m/z). Singly charged ions and ions with unassigned charge state were excluded

from triggering MS2 events. The normalised collision energy was set to 35%, the mass isolation window was set to 1.1 m/z, and one microscan was acquired for each spectrum.

The acquired raw files were converted to the mascot generic file (mgf) format using the msconvert tool (part of ProteoWizard, version 3.0.4624 (2013-6-3)). Converted files (mgf format) were searched by MASCOT (Matrix Sciences) against normal and reverse sequences (target decoy strategy) of the UniProt database of *A. baumannii* strains ATCC 19606 and ATCC 17978, and *P. aeruginosa* UCBPP-PA14, as well as commonly observed contaminants. The precursor ion tolerance was set to 20 ppm, and fragment ion tolerance was set to 0.02 Da. Full tryptic specificity was required (cleavage after lysine or arginine residues unless followed by proline), 3 missed cleavages were allowed, carbamidomethylation of cysteins (+57 Da) was set as fixed modification, and arginine (+10 Da), lysine (+8 Da), and oxidation of methionine (+16 Da) were set as variable modifications. For quantitative PRM experiments, the resolution of the orbitrap was set to 30,000 FWHM (at 200 m/z), and the fill time was set to 50 ms to reach a target value of 1e6 ions. Ion isolation window was set to 0.7 Th (isolation width), and the first mass was fixed to 100 Th. Each condition was analysed in biological triplicates. All raw files were imported into Spectrodive (Biognosys AG) for protein and peptide quantification.

## Supporting information

**S1 Fig. Anomalous data for the OprC$_{WT}$ Cu-SAD experiment.** (A, B) Copper anomalous maps (coloured magenta) contoured at 4 σ (carve = 30). Experimental density for one OprC protomer after density modification (but before model building) for (C) barrel and (D) N-terminal plug domain (map contoured at 1.5 σ, carve = 2.0). Ribbon is shown for orientation purposes. (E) X-ray fluorescence spectrum showing the copper-specific energy peak. Underlying data for panel E can be found in S1 Data.
(TIF)

**S2 Fig. ICP-MS data of OprC and mutant proteins.** (A) Metal occupancy of OprC and mutant proteins after incubation with 3 equivalents copper in the presence of 0.5 mM EDTA (approximately 50-fold excess) followed by analytical SEC and subsequent metal analysis by ICP-MS. (B) Metal occupancy of OprC WT after incubation with 3 or 10 Eq. of Cu or Zn. Underlying data for this figure can be found in S1 Data. EDTA, ethylenediamine tetra-acetic acid; ICP-MS, inductively coupled plasma mass spectrometry; SEC, size exclusion chromatography; WT, wild-type.
(TIF)

**S3 Fig. Structural changes upon copper binding suggest OprC is a bona fide TBDT.** (A, B) Cartoon superposition from the OM plane (A) and extracellular environment (B) of OprC$_{WT}$ (coloured green) and OprC$_{AA}$ (blue), indicating locations of loops L5, L8, and L11; copper is represented as a magenta sphere. The plug domains of OprC$_{WT}$ and OprC$_{AA}$ are coloured light green and light blue, respectively. (C) Superposition of N-terminal plug domains indicating the location of the TonB box, which is invisible in Cu-bound OprC$_{WT}$. Arrows indicate the missing density for Glu88-Pro94 in OprC$_{WT}$. (D) Surface slab representations from the OM plane, showing the presence of a solvent pocket in OprC$_{AA}$ that is generated by the absence of Met147 (arrow). For orientation purposes, the OprC$_{WT}$-bound copper is shown in both structures. (E) Side surface views showing the conformational changes of L8 and L11 (coloured yellow) as a result of copper binding. As in (D), the bound copper of OprC$_{WT}$ is shown in both proteins. OM, outer membrane; TBDT, TonB-dependent transporter; WT, wild-type.
(TIF)

**S4 Fig.** Stereo 3D representation of superposed OprC (coloured green) and OprC$_{AA}$ (blue). (A) Extracellular view showing loops 2, 4, 5, 6, 7, 8, 9, 10, 11 (L2, L4, L5, L6, L7, L8, L9, L10, L11). Conformational changes are observed for external loops L8 and L11. (B) Active site view illustrating superposed residues involved in metal coordination for WT OprC (green sticks) and OprC$_{AA}$ (blue sticks). Asn145 (N145) is also shown due to its role in shielding the active site. Oxygen atoms in amino acid residues are coloured red, nitrogens blue, and sulphurs yellow. The copper atom is represented as a magenta sphere. WT, wild-type.
(TIF)

**S5 Fig. Copper toxicity in *P. aeruginosa* cells overexpressing OprC.** Anaerobic growth of (A) pHERD30 (empty plasmid) and pHERD30-overexpressed (B) OprC$_{WT}$, (C) OprCC143A, and (D) OprC$_{AA}$ in PA14 *ΔoprC* was monitored during copper stress in rich media supplemented with 100 mM sodium nitrate. Overexpression was induced with 0.1% arabinose. Values indicate externally added copper in mM. Underlying data for this figure can be found in S1 Data.
(TIF)

**S6 Fig. Whole-cell metal content of PA14 WT and PA14 *ΔoprC* analysed via ICP-MS.** Cell-associated metal content was determined in cells grown in rich media supplemented with 100 mM sodium nitrate under both aerobic (left panels) and anaerobic conditions (right panels) without added copper. The 3 biological replicates have been plotted separately due to the different absolute metal contents. The right y-axis is for Fe content. Reported values are averages ± SD (*n* = 3). Underlying data for this figure can be found in S1 Data. ICP-MS, inductively coupled plasma mass spectrometry; WT, wild-type.
(TIF)

**S7 Fig. Amino acid sequence alignment for mature OprC sequences.** Shown are *P. aeruginosa* (uniprot ID G3XD89), NosA from *P. stutzeri* (uniprot ID Q00620), *Pseudomonas putida* (uniprot ID Q88DI7), *Pseudomonas syringae* (uniprot ID A0A085VGG7), *A. baumannii* (uniprot ID A0A0G4QL30), *S. enterica* (uniprot ID A0A505CFK3), *K. pneumonia* (uniprot ID A0A486MDQ0), *Serratia marcescens* (uniprot ID A0A221DQ80), and *Enterobacter cloacae* (uniprot ID A0A1S6XXV6), showing high conservation of the binding site residues Cys143 (highlighted in yellow), Met147 and Met325 (green), and His323 (cyan). Methionine track residues are depicted in red, and those located in the N-terminal plug are coloured magenta. The TonB box sequence is depicted in blue. The zinc transporter ZnuD from *Neisseria meningitides* (uniprot ID Q9JZN9) is shown for comparison. Numbering is for the full-length *P. aeruginosa* OprC sequence. Clustal scoring is indicated below the alignment.
(PDF)

**S8 Fig. Comparison of M147H and M325H binding site residues with pseudoazurin and plastocyanin.** Close-up views of copper binding site residues in (A) M147H, (B) M325H, (C) pseudoazurin (PDB ID 1PAZ), and (D) plastocyanin (PDB ID 4DPA). The bound form of copper is Cu(I). Distances between coordinating residues and metal (magenta) are shown. The table summarises distances between copper and coordinating residues as well as geometry.
(TIF)

**S9 Fig. Differences between OprC and the Zn-specific ZnuD.** (A, C) Cartoon representation comparing Cu-loaded OprC (coloured blue, copper atom shown as magenta sphere) and the locked version of ZnuD (coloured green, 2 cadmium atoms bound to low-affinity sites represented as grey spheres, zinc bound to the high-affinity site represented in orange; PDB ID 4RDR). The locked conformation of ZnuD shows low-affinity metal sites at external loops, at regions similar to the methionine track (L5) from OprC. (B, D) Transparent view of the

secondary structures illustrates the similar topological location for the high-affinity metal sites (distance of 2.8 Å).
(TIF)

**S1 Table. Data collection and refinement statistics for OprC variants with and without copper.**
(DOCX)

**S2 Table. Data collection and refinement statistics for OprC variants with silver.**
(DOCX)

**S3 Table. Data collection and refinement statistics for M147H and M325H variants.**
(DOCX)

**S1 Data. Source data underlying the main figures and supporting information figures except Fig 4B.**
(XLSX)

**S2 Data. Uncropped original gel image of Fig 4B.**
(TIF)

## Acknowledgments

We would like to acknowledge Scott Sucoloski, Jennifer Hoover, and Josh West (Glaxo Smith Kline) for providing proteomics samples. We thank Bastien Belzunces, Chris Skylaris, and Syma Khalid (University of Southampton) for exploratory quantum chemical calculations. We also thank Kevin Waldron (Newcastle University) for useful discussions and for carrying out initial ICP-MS analyses. We also thank Deenah Osman and Nigel Robinson (Durham University) for ICP-MS analyses and helpful discussions, supported by awards BB/L009226/1 and BB/R002118/1 from the BBSRC. We are indebted to the Diamond Light Source for beam time (proposals mx9948, mx13587, and mx18598) and beamline assistance. MS acknowledges the EPSRC National (UK) EPR Research Facility and Service for use of the EPR spectrometers. MS and BvdB thank Luisa Ciano for the useful early stage EPR discussions.

## Author Contributions

**Conceptualization:** Bert van den Berg.

**Funding acquisition:** Bert van den Berg.

**Investigation:** Satya Prathyusha Bhamidimarri, Tessa R. Young, Muralidharan Shanmugam, Sandra Soderholm, Arnaud Baslé, Bert van den Berg.

**Resources:** Arnaud Baslé.

**Software:** Arnaud Baslé.

**Supervision:** Dirk Bumann, Bert van den Berg.

**Validation:** Satya Prathyusha Bhamidimarri, Muralidharan Shanmugam.

**Visualization:** Satya Prathyusha Bhamidimarri, Bert van den Berg.

**Writing – original draft:** Satya Prathyusha Bhamidimarri, Bert van den Berg.

**Writing – review & editing:** Satya Prathyusha Bhamidimarri, Tessa R. Young, Bert van den Berg.

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
