## [Editor Report · Decision Letter 0]

19 Aug 2021

Dear Dr. Van Den Berg, 

Thank you for submitting your manuscript entitled "Acquisition of ionic copper by a bacterial outer membrane protein" for consideration as a Research Article by PLOS Biology.

Your manuscript has now been evaluated by the PLOS Biology editorial staff, as well as by an academic editor with relevant expertise, and I am writing to let you know that we would like to send your submission out for external peer review.

Please re-submit your manuscript within two working days, i.e. by Aug 21 2021 11:59PM.

Kind regards,

Paula 

---

Paula Jauregui, PhD

Associate Editor

PLOS Biology

---

## [Decision Letter · Decision Letter 1]

30 Sep 2021

Dear Dr Van Den Berg,

Thank you for submitting your manuscript entitled "Acquisition of ionic copper by a bacterial outer membrane protein" for consideration as a Short Report by PLOS Biology. Please accept my apologies for the delay in getting back to you with our decision. As with all papers reviewed by the journal, yours was evaluated by the PLOS Biology editors as well as by an Academic Editor with relevant expertise and by two independent reviewers. 

The reviews are attached below. As you can see, the reviewers appreciate the attention to an important topic and note that the study is well-executed. Based on the reviews, we will probably accept this manuscript for publication, provided you satisfactorily address the remaining points raised by the reviewers. This includes a couple of reporting concerns as well as including additional discussions about the methionine track binding model.

In addition, we ask that you please address the following editorial and other policy-related requests that I have provided below:

A) We would to suggest the following modification to the title, to make it more compelling for our broad readership:

'Acquisition of ionic copper by the bacterial outer membrane protein OrpC through a novel binding site'

B) You may be aware of the PLOS Data Policy, which requires that all data be made available without restriction: http://journals.plos.org/plosbiology/s/data-availability. For more information, please also see this editorial: http://dx.doi.org/10.1371/journal.pbio.1001797

- Supplementary files (e.g., excel). Please ensure that all data files are uploaded as 'Supporting Information' and are invariably referred to (in the manuscript, figure legends, and the Description field when uploading your files) using the following format verbatim: S1 Data, S2 Data, etc. Multiple panels of a single or even several figures can be included as multiple sheets in one excel file that is saved using exactly the following convention: S1_Data.xlsx (using an underscore).

- Deposition in a publicly available repository. Please also provide the accession code or a reviewer link so that we may view your data before publication.

Regardless of the method selected, please ensure that you provide the individual numerical values that underlie the summary data for the following Figures, as they are essential for readers to assess your analysis and to reproduce it:

Figure 2A-F, 4A, S1E, S2A-B, S5A-D, S6

C) Please also ensure that each of the relevant figure legends in your manuscript include information on *WHERE THE UNDERLYING DATA CAN BE FOUND*, and ensure your supplemental data file/s has a legend

D) Please ensure that your Data Statement in the submission system accurately describes where your data can be found and is in final format, as it will be published as written there. 

E) We require the original, uncropped and minimally adjusted images supporting the gel image reported in Figure 4B. We will require these files before a manuscript can be accepted so please prepare and upload them now. Please carefully read our guidelines for how to prepare and upload this data: https://journals.plos.org/plosbiology/s/figures#loc-blot-and-gel-reporting-requirements

F) Please also provide a blurb which (if accepted) will be included in our weekly and monthly Electronic Table of Contents, sent out to readers of PLOS Biology, and may be used to promote your article in social media. The blurb should be about 30-40 words long and is subject to editorial changes. It should, without exaggeration, entice people to read your manuscript. It should not be redundant with the title and should not contain acronyms or abbreviations. For examples, view our author guidelines: https://journals.plos.org/plosbiology/s/revising-your-manuscript#loc-blurb

----------

We expect to receive your revised manuscript within two weeks. 

*Published Peer Review History*

*Early Version*

Sincerely,

Richard

Richard Hodge, PhD

Associate Editor, PLOS Biology

rhodge@plos.org

On behalf of:

Paula Jauregui, PhD

Associate Editor, PLOS Biology

Reviewer remarks:

Reviewer #1: Summary: The authors determined the structure of OprC, a TonB-dependent transporter from P. aeruginosa, with and without metal bound. The copper binding site for OprC is unique with the motif CxxxM-HxM motif binding Cu(II) in the structure. The authors also see a "methionine track" leading Cu(I) to the principal metal binding site. Using Ag(I) as a reduced copper mimic, they solved the structure of OprC showing Ag(I) density along the "methionine track". The authors also used EPR and ICP-MS to characterize the specificity to which OprC binds copper. Mutation studies of the residues at the principal binding site further show the specificity of OprC to binding Cu(II) at the binding site, while Cu(I) can be bound to the "methionine track"

Comments:

Figure 1 - It would be helpful if the colors of the protein structures were unique to each structure. Changing the colors of the structures would allow the reader to more easily follow along with the authors' comparison of the structures. Specifically, this will help with 1E and 1F when the authors discuss the differences in the TonB box. 

Figure 2 - Please include a figure legend for the graphs in Figure 2A-C. This will help the readers follow the data better without having the read the figure captions to understand which samples are being compared. 

The section "OprC is abundant in P. aeruginosa during infection" could be better discussed to explain why the authors performed these experiments.

Reviewer #2: This manuscript reports X-ray crystal structures of wild type and mutant OprC, an outer membrane copper importer, in the absence and presence of copper or silver, and characterized metal binding via ICP-MS and EPR. This is a very interesting and thoughtful study that provides detailed structural and mechanistic insights into copper transport by OprC. This is an excellent study shedding light on bacterial copper acquisition. Addressing the following concerns would further increase the significance of this beautiful work. 

1. Page 9, the OprC methionine track. The authors used silver co-crystal structures to identify 3 distinct binding sites, Ag1, Ag2 and Ag3. Interestingly, 2 binding sites (Ag2 and Ag3) are clearly located in the methionine track. What is the distance between sites Ag2 and Ag3? Could the methionine track simultaneously bind Ag2 and Ag3? It appears that each site is constructed by 3 methionine residues (which has been seen in other Cu+ transporters) from Figure 3A, with M339 and M343 being shared between these two sites. Please discuss these possibilities.

2. Transport pathway and mechanism. Very interestingly, Figure 1J reveals the methionine network potentially involved in Cu transport across the outer membrane. The extracellular portion of the Cu path is convincing. Some methionine residues are present in the plug domain. Could the authors outline the Cu path across the membrane? Does the plug domain need to swing out to the periplasm to allow Cu pass through the membrane?

---

## [Editor Report · Decision Letter 2]

18 Oct 2021

Dear Dr. Van Den Berg,

On behalf of my colleagues and the Academic Editor, Ann Stock, I am pleased to say that we can in principle offer to publish your Short Reports "Acquisition of ionic copper by the bacterial outer membrane protein OprC through a novel binding site" in PLOS Biology, provided you address any remaining formatting and reporting issues. These will be detailed in an email that will follow this letter and that you will usually receive within 2-3 business days, during which time no action is required from you. Please note that we will not be able to formally accept your manuscript and schedule it for publication until you have made the required changes. Please review your reference list to ensure that it is complete and correct, as we noticed that page numbers are missing for the newly added reference (ref # 52).

PRESS

Sincerely, 

Paula 

---

Paula Jauregui, PhD 

Associate Editor 

PLOS Biology
